# Thyroid Hormone Receptor β Knockdown Reduces Suppression of Progestins by Activating the mTOR Pathway in Endometrial Cancer Cells

**DOI:** 10.3390/ijms232012517

**Published:** 2022-10-19

**Authors:** Bingtao Ren, Jieyun Zhou, Yingyi Hu, Ruihua Zhong, Qiaoying Lv, Shuwu Xie, Guoting Li, Bingyi Yang, Xiaojun Chen, Yan Zhu

**Affiliations:** 1School of Pharmacy, Fudan University, Shanghai 200032, China; 2Lab of Reproductive Pharmacology, NHC Key Lab of Reproduction Regulation, Shanghai Institute for Biomedical and Pharmaceutical Technologies, Fudan University, Shanghai 200032, China; 3Department of Gynecology, Shanghai Key Laboratory of Female Reproductive Endocrine Related Diseases, Obstetrics and Gynecology Hospital, Fudan University, Shanghai 200011, China

**Keywords:** thyroid hormone receptor, endometrial cancer, medroxyprogesterone acetate, nomegestrol acetate, progestin resistance, mTOR signaling pathway

## Abstract

Progestin resistance is a major obstacle to conservative therapy in patients with endometrial cancer (EC) and endometrial atypical hyperplasia (EAH). However, the related inducing factor is yet unclear. In this study, thyroid hormone and its receptor α (TRα) and β (TRβ) of patients were assayed. THRB-silenced RL95-2 and KLE EC cells were cultured to investigate the response of progestins. Transcriptomics and Western blotting were performed to investigate the changes in signaling pathways. We found that THRB, rather than THRA, knockdown promoted the viability and motilities of RL95-2 cells but not KLE cells. The suppressive effect of progestins on cell growth and motility significantly decreased in THRB-silenced RL95-2 cells. Multiple proliferation-related signaling pathways were enriched, and the activities of mammalian targets of rapamycin (mTOR)/4e-binding protein 1 (4EBP1)/eukaryotic translation initiation factor 4G (eIF4G) rather than phosphorylated protein kinase B (Akt) were remarkably boosted. Progestin treatment enhanced the effects, and the augmentation was partially abated on supplementation with T3. In THRB-knockdown KLE cells, the progestins-activated partial signaling pathway expression (either mTOR or eIF4G), and supplementation with T3 did not induce noticeable alterations. The serum levels of triiodothyronine (T3) were significantly lower in patients with EC compared with healthy women. A strong expression of TRβ was observed in most patients with EC and EAH sensitive to progestin treatment. In contrast, TRα positive expression was detected in less than half of the patients sensitive to progestin therapy. In conclusion, THRB knockdown enhanced the viability and motility of type I EC cells and attenuated the suppressive effects of progestins by activating the mTOR-4EBP1/eIF4G pathway. Lower expression of THRB is likely correlated with progesterone resistance.

## 1. Introduction

Endometrial cancer (EC) originates in the cellular layer of the endometrium and is one of the most common malignancies of the female reproductive tract [1]. Although most patients are diagnosed after menopause, the incidence of EC has gradually increased in fertile women [2,3]. EC is generally classified as type I or type II based on its pathological and molecular features. Type I EC positively responds to estrogen receptors (ERs) and progesterone receptors (PRs) and accounts for 70–80% of EC. Type II EC has a negative or weakly positive response to ERs and PRs, and accounts for 20–30% of EC [4,5]. Standard surgical treatments, including hysterectomy and bilateral salpingo-oophorectomy, are not suitable for all patients with EC, especially for young patients with type I EC who wish to preserve their fertility. For these patients, progestin treatments are generally allowed after a rigorous evaluation. At present, the medication approved officially for EC therapy is medroxyprogesterone acetate (MPA) [6]. However, some patients experience progestin resistance during the therapy, failing treatment for approximately 30% of patients [7,8,9]. MPA has an initial response rate of 55–100% [10,11], but an overall response rate of only 35% during the therapy [12]. Additionally, the complete response rate for the patients taking low-dose (200 mg/day) and high-dose (1000 mg/day) MPA has been shown to be 17% and 9%, respectively [13,14], demonstrating an opposite dose–response relationship. These pieces of evidence indicate the existence of progestin resistance. The down-regulation of PR and the abnormal activation of phosphatidylinositol-3-hydroxykinase (PI3K)/phosphorylated protein kinase B (Akt) are generally considered as the main factors inducing progestin resistance in EC [15,16]. However, the theory does not explain why some patients with ER- and PR-positive expression do not respond to progestins while others with ER- and PR-negative expression respond to hormonal treatment [8]. Accordingly, it is plausible to speculate that unknown factors are associated with progestin resistance.

Nuclear thyroid hormone receptors (TRs) have two isoforms: thyroid hormone receptor α (TRα) and thyroid hormone receptor β (TRβ), encoded by THRA and THRB, respectively, mediating the effects of thyroid hormones (THs) [17]. The state of transcriptional regulation for the target genes of TRs depends on triiodothyronine (T3), which is the ligand and agonist of TRs [18]. Studies showed that hypothyroidism might be a risk factor for many tumors, including liver, breast, and thyroid [19,20,21,22], and the expression of TRα and TRβ could be one of the factors affecting cancer progression [23,24,25,26,27]. Recent evidence shows that hypothyroidism may also correlate with the occurrence of EC because it is one of the comorbidities in patients with EC [21]. Thyroid dysfunction occurs in about 8.2% of patients with EC. Its incidence ranks second to that of metabolic syndrome [28] but the relationship between thyroid dysfunction and EC has not yet been elucidated. Earlier, it was reported that high doses of MPA could increase the uptake of T3 during the therapy for EC and renal carcinoma [29]. This implies that MPA promotes the binding of more thyroxine-binding globulin to T4 or T3. Nevertheless, how THs and TRs influence the progression of EC and the therapeutic outcome of progestins are currently unknown.

The presented study focused on the role of THRB in the growth of EC cells and its impact on Akt/ mammalian target of rapamycin (mTOR) pathway in the absence and presence of progestins. Both MPA and nomegestrol acetate (NOMAc) were tested since NOMAc, one of the fourth-generation progestins, has demonstrated stronger chemical properties than MPA and exhibited a suppressive effect on mTOR and its downstream signaling pathway in previous experiments [30].

## 2. Results

### 2.1. Lower Serum FT3 Levels Were Associated with EC

The analysis of the serum data revealed that the serum levels of FT3, FT4, and thyroid-stimulating hormone (TSH) in healthy women of different ages did not change significantly (*p* > 0.05, Figure 1A). However, the levels of FT3 rather than of T4 and TSH were significantly lower in patients with EC than in healthy women (*p* = 0.031, Figure 1B). It indicated that lower FT3 levels might be significantly correlated with EC.

Immunohistochemistry (IHC) was used to detect the expression of TRα and TRβ in progestin-sensitive and progestin-insensitive EAH and EC tissues. The staining of TRα and TRβ proteins was mainly localized in the cytoplasm and some localized in the nucleus, showing yellow or brown colors. The histoscore of TRα was 42% (5/12) in progestin-sensitive tissues and 71% (5/7) in progestin-insensitive tissues. The ratio of the positive area for TRα was not significantly different between progestin-sensitive (14.13 ± 3.75%) and progestin-insensitive (17.81 ± 2.33%) tissues (*p* > 0.05, Figure 1C,D). In contrast, the histoscore of TRβ was 100% (13/13) in progestin-sensitive tissues, and the strong intensity of immunostaining rate was 77% (10/13). However, the histoscore of TRβ was 85% (6/7) in the progestin-insensitive endometrium, and the strong intensity of immunostaining rate was 33% (2/6). The ratio of positive area for TRβ was higher in progestin-sensitive (37.23 ± 2.28%) tissues than that in progestin-insensitive (19.27 ± 2.58%) tissues (*p* < 0.05, Figure 1C,D).

Consistently, both TRα and TRβ were found to be abundantly expressed in both RL95-2 and KLE cells, as shown in Figure 1E.

### 2.2. Effects of Progestins on the Viability of EC Cells

The inhibitory effects of two progestins, MPA and NOMAc, and their combination with T3 on the viability of two types of EC cell lines, RL95-2 and KLE, were evaluated. As shown in Figure 2A, the treatment with both MPA and NOMAc in the concentration range from 1–100 μM for 48 h significantly inhibited the growth of RL95-2 cells in a concentration-dependent manner (*p* < 0.05), with the IC50 (50% inhibitory concentration) values being 52.25 μM and 36.81 μM, respectively (Table 1). In KLE cells, both of the progestins suppressed the viability of the cells at concentrations of more than 30 μM (Figure 2B), with the calculated IC50 value of NOMAc and MPA being 281.2 μM and greater than 400 μM, respectively (Table 1).

When the cells were treated with T3 alone at a concentration of 10 or 100 nM, no distinct changes were observed in the growth of both RL95-2 and KLE cells compared with the control cells (*p* > 0.05, Figure 2C−F). However, the viability of RL95-2 cells markedly increased when T3 and 30 μM MPA or NOMAc were simultaneously added to the cells, and the inhibitory effect of progestins on RL95-2 cells significantly decreased compared with that of the progestin treatment alone (*p* < 0.05, Figure 2C,E). Conversely, T3 combined with MPA or NOMAc treatment did not significantly change the suppressive effect of the progestins in KLE cells (*p* > 0.05, Figure 2D,F).

### 2.3. Knockdown of THRB Not THRA Promoted the Growth of RL95-2 Cells, Which Was Further Enhanced by Progestins

Using qRT-PCR, both THRA and THRB were found to be highly expressed in both RL95-2 and KLE cells with the δCT value between 6 and 8. Both genes were effectively silenced after treatment with siRNA reagents for 48 h, and their protein expression was less detectable after 96 h, as shown in Figure 3A–C.

The viability of THRA-silenced RL95-2 cells did not change much compared with that of controls (si-Ctrl) (without THRA or THRB knockdown and just treated with si-RNA reagents) (*p* > 0.05, Figure 3D), treatment with both progestins did not remarkably alter the growth of the cells (*p* > 0.05, Figure 3D–F).

In THRB-silenced RL95-2, however, the viability of the cells was enhanced in a time-dependent manner, and a significant increase was observed at 72 and 96 h after transfection compared with that of si-Ctrl cells (*p* < 0.05, Figure 3D). Moreover, the growth abilities of the cells were significantly enhanced after MPA or NOMAc treatment for 48 h at the concentration of 30 μM compared with the control cells (*p* < 0.05, Figure 3E,F). The supplementation with T3, the agonist of TRα and TRβ, abolished the significant difference in the viabilities between the THRB-silenced RL95-2 cells and the control cells after treatment with the progestins; while T3 treatment alone did not alter the growth of the THRB-silenced RL95-2 cells (*p* > 0.05, Figure 3E,F), similar to that in wild-type RL95-2 cells. It suggests that THRB, rather than THRA, likely plays a role in regulating the growth of type I EC cells and interferes with the action of the progestins.

The knockdown of THRA or THRB did not remarkably affect the growth of KLE cells (*p* > 0.05, Figure 3G). Additionally, the progestin treatment did not significantly change the growth of THRA- and THRB-silenced KLE cells compared with that of si-Ctrl cells (*p* > 0.05, Figure 3H,I). Additionally, combining the progestins with T3 did not affect the viability of the THRA-silenced and THRB-silenced KLE cells compared with the treatment with the progestins alone (*p* > 0.05, Figure 3H,I).

### 2.4. Knockdown of THRB, Not THRA, Promoted the Motility of RL95-2 Cells, Which Was Further Enhanced by Progestins

Using the transwell migration assay, we found that the knockdown of THRB, but not THRA, remarkably increased the cell’s migratory ability in RL95-2 cells (*p* < 0.05, Figure 4A). In THRB-silenced RL95-2 cells, MPA or NOMAc treatment significantly decreased the number of migrating cells compared with the cells treated with DMSO. In MPA- or NOMAc-treated RL95-2 cells, however, THRB silencing increased the number of migrating cells by 15.54% (*p* > 0.05, Figure 4A) and 43.85% (*p* < 0.05, Figure 4A), respectively, compared with that of si-Ctrl cells.

Moreover, the number of invading cells significantly decreased after the knockdown of THRB (*p* < 0.05, Figure 4B), not THRA (*p* > 0.05, Figure 4B), in DMSO-treated RL95-2 cells. In THRB-silenced RL95-2 cells, MPA and NOMAc treatment significantly boosted cell invasion compared with the treatment with DMSO (*p* < 0.05, Figure 4B). In *THRA*-silenced RL95-2 cells, on the contrary, MPA and NOMAc treatment suppressed the invasive capabilities of the cells compared with the treatment with DMSO (*p* < 0.05, Figure 4B). It suggests that the knockdown of THRB, not THRA, promotes cell migration, and progestins facilitate cell invasion in THRB-silenced RL95-2 cells.

In KLE cells without THRA or THRB silencing, the cell migration did not change but the invasion was suppressed after treatment with MPA or NOMAc compared with control cells treated with DMSO (*p* < 0.05, Figure 4D). In THRA-silenced KLE cells, the cell migration and invasive abilities did not significantly decrease after treatment with NOMAc (*p* > 0.05, Figure 4C,D), but treatment with MPA significantly inhibited cell migration compared with non-silenced cells (*p* < 0.05, Figure 4C). NOMAc decreased the number of migrating cells (*p* < 0.05, Figure 4D) but affected cell invasion less in THRB-silenced KLE cells compared with control cells treated with DMSO or non-silenced cells. It suggests that MPA and NOMAc treatment reduces the migratory abilities in either THRA- or THRB-silenced in KLE cells, differently from that in RL95-2 cells.

### 2.5. Identification of DEGs and Associated Signaling Pathways in Progestin-Treated THRB-Silenced RL95-2 Cells

Considering the facilitative effects of progestins on the growth and motility of THRB-knockdown RL95-2 cells, we further explored the signaling pathway possibly involved. The flow chart of the experiment is shown in Figure 5A. A total of 570 DEGs were obtained in THRB-knockdown RL95-2 cells compared with the negative si-Ctrl cells, involving 178 up-regulated and 392 down-regulated DEGs (Figure 5B–D). Moreover, 4500 and 4377 DEGs were obtained in MPA- and NOMAc-treated THRB-knockdown RL95-2 cells, respectively, compared with the cells treated with DMSO, which included 2776 up-regulated and 1724 down-regulated DEGs in MPA treatment groups (Figure 5B,E), and 3028 up-regulated and 1349 down-regulated DEGs in NOMAc treatment groups, respectively, as shown in the heatmap (Figure 5B,F); the Venn diagram shows the overlap of DEGs (Figure 5C,F). The DEGs were subsequently enriched by KEGG pathway analysis. Most of the DEGs were found to be involved in signaling pathways correlated with cell proliferation, apoptosis, and immune regulation, including MPAK, Ras, PI3K/Akt, mTOR, ErbB, and IL-17 signaling pathways (Figure 5G–I). Especially, the heatmaps of DEGs associated with thyroid hormone, PI3K-Akt, and mTOR signaling are shown in Appendix A. The most important DEGs in the pathways included DIO2, ITGA3, ESR1 and ITGAV in the thyroid hormone signaling pathway, and PIK3R2, PIK3CA and AKT3 in the PI3K-Akt signaling pathway and RICTOR, LAMTOR4, LAMTOR1 and EIF4E2 in the mTOR signaling pathway. The DEGs between si-Ctrl-RL95-2 and si-THRB-RL95-2 cells and si-THRB-DMSO-RL95-2 and si-THRB-MPA or NOMAc RL95-2 cells are listed in Appendix A.

### 2.6. Progestins Activated the mTOR-4EBP1/eIF4G Signaling Pathway in THRB-Silenced RL95-2 EC Cells

The activity and protein levels of Akt and its downstream mTOR/4e-binding protein 1 (4EBP1)/eukaryotic translation initiation factor 4G (eIF4G) were further investigated since the signaling pathway was enriched in THRB-knockdown RL95-2 cells and progestin treatment groups.

In RL95-2 cells, 30 μM MPA significantly increased the protein levels of phosphorylated Akt (p-Akt) and eIF4G (p-eIF4G) (*p* < 0.05, Figure 6A) compared with control cells and did not affect the levels of phosphorylated 4EBP1 (p-4EBP1) and mTOR (p-mTOR) (*p* > 0.05, Figure 6A). Conversely, 30 μM NOMAc significantly decreased the levels of p-mTOR (*p* < 0.05, Figure 6(A-2)) but did not affect the levels of p-Akt, p-4EBP1 and p-eIF4G (*p* > 0.05, Figure 6A). This was consistent with our previous findings [30]. It suggests that NOMAc exhibits superiority over MPA in suppressing the pathway of Akt/mTOR/p-4EBP1/p-eIF4G. In addition, T3 treatment alone remarkably increased the protein level of p-eIF4G but did not affect the levels of p-Akt, p-mTOR, and p-4EBP1 (*p* < 0.05, Figure 6A). When the progestins were combined with T3, the levels of p-Akt and p-mTOR were not much different from those for the progestins treatment alone, but a significant decline was observed in the levels of p-4EBP1 and p-eIF4G in the NOMAc or MPA treatment groups, respectively (*p* < 0.05, Figure 6A). It suggests that T3 hardly boosts the activity of the mTOR signaling pathway in RL95-2 cells when combined with the progestins.

However, the protein levels of p-mTOR, p-4EBP1, and p-eIF4G significantly increased but the levels of p-Akt decreased in THRB-silenced RL95-2 cells compared with that of si-Ctrl cells (*p* < 0.05, Figure 6(B-1–B-4)), and the same effects were also observed in the cells treated with the progestins or T3 alone (*p* < 0.05, Figure 6(B-1–B-4)). When the cells were treated with NOMAc and T3 together, the expression of p-mTOR and p-4EBP1 moderately declined and the significant differences between the si-Ctrl and si-THRB groups were abolished (*p* > 0.05, Figure 6(B-2,B-3)). When the cells were treated with MPA and T3 together, the levels of p-mTOR declined and the significant differences between si-Ctrl and si-THRB also decreased (*p* > 0.05, Figure 6(B-2)), but the activity of p-eIF4G remarkably enhanced (*p* > 0.05, Figure 6(B-4)). It meant that the facilitative effects of progestins on mTOR/ 4EBP1/eIF4G were based on THRB silencing and adding T3 could partially antagonize the boosting effect of the progestins in the THRB-silenced RL95-2 cells.

In KLE cells, T3 did not affect the signaling pathway. MPA significantly increased the levels of p-eIF4G but decreased the levels of p-mTOR and p-4EBP1 compared with control cells (*p* < 0.05, Figure 6C), similar to those in RL95-2 cells; while NOMAc treatment alone did not affect the protein expression of p-Akt, p-mTOR, p-4EBP1 and p-eIF4G (*p* > 0.05, Figure 6C). It suggests that NOMAc hardly boosts the activity of the Akt/mTOR pathway in KLE cells. In THRB-silenced KLE cells, THRB knockdown did not affect the levels of p-Akt but inhibited the expression of p-mTOR and increased the expression of p-4EBP1 and p-eIF4G compared with the control cells without silencing (*p* < 0.05, Figure 6D), and similar effects were observed in T3 treatment cells. MPA treatment alone increased the levels of p-mTOR (*p* < 0.05, Figure 6(D-2)) but did not significantly affect the levels of p-Akt, p-4EBP1, and p-eIF4G (*p* > 0.05, Figure 6(D-1,D-3,D-4)). When combined with T3, the pronounced elevating effect of MPA on p-mTOR expression abrogated (*p* > 0.05, Figure 6(D-2)) but a statistically significant increase was observed in the levels of p-eIF4G compared with control cells (*p* < 0.05, Figure 6(D-4)). Conversely, NOMAc treatment alone and its combination with T3 did not affect the levels of p-Akt or p-mTOR but decreased the levels of p-4EBP1 and increased the levels of p-eIF4G (Figure 6D). As a result, both progestins exhibited similar promoting effects on the activity of eIF4G in both THRB-silenced RL95-2 and KLE cells; however, the effects on the activity of Akt and mTOR/4EBP1 were different.

## 3. Discussion

The present study was novel in demonstrating that THRB knockdown promoted the growth and motility of RL95-2 EC cells and attenuated the suppressive effect of the progestins via enhancing the activity of the mTOR-4EBP1/eIF4G pathway. Thus, progestins facilitated the growth of type I EC cells in the absence of THRB. We found lower serum levels of T3 in patients with EC compared with healthy women. Moreover, weaker expressions of TRβ were observed in the endometrium of patients with EC or EAH insensitive to progestin therapy compared with those sensitive to the therapy. Taken together, it was plausible to presume that the lower expression of THRB/TRβ was likely correlated with progestin resistance in EC therapy.

Human type I EC is generally described as hormone receptor-positive and sensitive to progestin therapy and type II EC is described as hormone receptor-negative and insensitive to progestin therapy. Accordingly, human-originated RL95-2 cells are defined as type I EC cells and KLE as type II EC cells, which are characterized by positive and negative expression of hormone receptors, respectively. In our previous study, the protein expression of ERα and PR was observed in RL95-2 cells but not in KLE cells, and the expression of p53 was detected in KLE cells but not in RL95-2 cells [30]. In this study, therefore, we used both RL95-2 and KLE cells to investigate the effect of THRB on progestin treatment and found that RL95-2 cells were more sensitive to MPA and NOMAc than KLE cells, which was consistent with the previous findings [30].

THRB has been reported to act as a transcription-suppressive factor in response to the changes in THs levels, lying at the crossroad of many cellular signaling pathways and playing a critical role in maintaining normal cell characteristics and tumor progression in thyroid cancer [25]. THRB may act as a tumor suppressor in solid tumors, such as breast, hepatocyte, and thyroid, by inducing apoptosis and reducing the cell renewal capacity to restrain the growth of tumors [25,26]. The present study found that both TRα and TRβ were expressed in RL95-2 and KLE cells; however, it was THRB knockdown, rather than THRA, boosting the proliferation and motility in RL95-2 cells. It indicated that THRB exerted a suppressive effect on cell proliferation, when THRB was knocked down, the suppressive effect was attenuated.

The Akt-mTOR pathway is generally considered to play a role in the proliferation of tumor cells and is involved in progestin resistance [16,31]. Akt inhibitor has been used in treating ER/PR-positive breast cancer to disrupt the function of the PI3K/Akt/mTOR pathway and alleviate endocrine resistance [16,32]. The mTOR inhibitor has also been tested as an EC therapy in phase II clinical trials [33,34]. We previously found that activating the mTOR-4EBP1/eIF4G pathway could promote proliferation and inhibit apoptosis in RL95-2 and HEC-1A EC cells [30]. The present study found that the growth and the motility of cells were enhanced, and mTOR and its downstream 4EBP1/eIF4G signaling pathway were significantly activated in THRB-silenced RL95-2 cells. In contrast, not the whole of the signaling pathway was activated in THRB-silenced KLE cells, which might result in little impact on the growth and motility of the cells. It is plausible to presume that *THRB* likely plays a more critical role in regulating the growth of RL95-2 cells than KLE cells, and its action is correlated with the mTOR signaling pathway.

Progestin resistance caused by EC cells insensitive to the treatment of progestins has been a major obstacle for patients who wish to preserve fertility, and the mechanism involves the activation of the PI3K/Akt pathway and abnormal proliferation of tumor cells [31,35,36]. Previously, we found that NOMAc effectively restrained the growth of RL95-2 and HEC-1A cells by suppressing the activity of mTOR-4EBP1/eIF4G. In this study, we further found that several proliferation-related signaling pathways, including PI3K/Akt, mTOR, Ras, MAPK, and TP53, were markedly enriched in the progestin treatment of THRB-silenced RL95-2 cells using the transcriptomic assay. Accordingly, we focused on Akt and its downstream mTOR signaling pathway to explore how progestins induced cell growth. We found that both MPA and NOMAc suppressed the activity of Akt but significantly facilitated the activity of mTOR and its substrates 4EBP1 and eIF4G in THRB-silenced RL95-2 cells. Additionally, the supplementation of T3 partially abrogated the pronounced boosting effect of the progestins on mTOR and 4EBP1. In THRB-silenced KLE cells, the progestins did not affect the activity of Akt but increased the activity of mTOR or eIF4G. It suggests that MPA and NOMAc activate mTOR/4EBP1/eIF4G rather than Akt in the THRB-silenced EC cells. In view of the remarkable promoting effects of both progestins on cell growth and motility in the THRB-silenced RL95-2 cells, it was plausible to presume that the down-regulation of THRB might be one of the critical factors associated with progestin resistance in type I EC cells. The underlying mechanism was probably via activating mTOR and its downstream 4EBP1/eIF4G signaling pathway rather than upstream Akt, but more related studies are warranted in the future.

Activating Akt, the upstream signaling of mTOR was partially ascribed to MPA-induced resistance in EC and breast cancer [16,31]. Consistently, we also observed that MPA enhanced the activity of Akt and eIF4G in both RL95-2 and KLE cells. In contrast, NOMAc facilitated the activity of Akt in RL95-2 cells but not in KLE cells. The results could partially expound the reason why NOMAc exhibited stronger inhibition than MPA on the growth of KLE cells. In THRB-silenced EC cells, both MPA and NOMAc demonstrated similar boosting effects on p-mTOR and p-eIF4G, except that NOMAc notably decreased the activity of 4EBP1 in THRB-silenced KLE cells. It suggests that MPA and NOMAc display subtly different effects on the proliferation-related signaling pathways in both wild-type and THRB-silenced EC cells.

Consistent with the findings of the cell experiments, we found a disparity in the protein expression of TRα and TRβ in patients sensitive or insensitive to progestin therapy. Both endometrium tissues were detected in the study because the number of patients with EAH was more than that of patients with EC. Notably, the expression of TRβ in progestin-sensitive EAH and EC tissues was significantly stronger than that in progestin-insensitive tissues. In contrast, no significant difference was found in the expression of TRα between progestin-sensitive and progestin-insensitive tissues. It suggested that the expression of TRβ, rather than TRα, might influence the effect of progestins in treating of EAH/EC, and a stronger expression of TRβ was more likely correlated with the effective therapy using progestins.

Generally, hypothyroidism is diagnosed based on changes in the levels of TSH, total thyroxine (TT4), and free thyroxine (FT4) [21]. However, we found that the serum levels of FT3, rather than FT4 and TSH, significantly declined in patients with EC compared with healthy women in this study. Although low-T3 syndrome has been observed in many patients with chronic disease or cancer, including type II diabetes mellitus [37] and chronic lymphocytic leukemia [38], whether the low levels of T3 are correlated with the occurrence of EC is unclear. Nevertheless, the result provides evidence that a relationship that exists between EC and abnormal thyroid systemic function. More clinical investigations are warranted in the future.

We also investigated the effect of T3 and found that both 10 and 100 nM T3 demonstrated similar effects on the EC cells and did not influence the growth of the two types of EC cells but antagonized the suppressive effects of progestins in RL95-2 cells rather than in KLE cells. Therefore, 100 nM T3 was used as an agonist of THRB for subsequent investigation to distinguish the effects induced by exogenous T3, rather than endogenous T3, in FBS because normal FBS (possibly containing THs), rather than hormone-depleted serum, was used in the present study to avoid more severe injury to the cells in the presence of transfection reagents. In THRB-silenced RL95-2 cells, we found that T3 did not change the viability of the cells but abolished cell proliferation induced by progestins, which might be ascribed to the inconsistent or even inversed regulation of T3 on the activity of Akt and mTOR/4EBP1/eIF4G. T3 treatment significantly enhanced the activity of mTOR-4EBP1/eIF4G but not of Akt; while a combination with the progestins increased the activity of either p-4EBP1 or p-eIF4G and markedly decreased the activity Akt. As a result, the proliferative effects of progestins were mildly abolished. These pieces of evidence indicate that T3 supplement likely restored the suppressive effects of the progestins in THRB-silenced RL95-2 cells but not in THRB-expressing cells. Moreover, despite the fact that T3 and progestins demonstrated facilitative or suppressive effects on the signaling of Akt/mTOR-4EBP1/eIF4G in THRB-silenced EC cells, no significant difference was observed between DMSO and T3 or progestins treatment groups. It suggests that the alteration of the signaling pathway arose from the knockdown of THRB rather than induced by T3 or progestins themselves. Taken together, lower expression of THRB is likely one of the crucial factors causing progestin resistance.

## 4. Conclusions and Limitations

In conclusion, we demonstrated a remarkable link between TRβ knockdown and cell proliferation in type I RL95-2 EC cells. The silencing of THRB would impair the suppressive effects of progestins and activate the mTOR-4EBP1/eIF4G pathway rather than Akt signaling, which was likely one of the factors causing progesterone resistance in the therapy of type I EC. The status of THRB may play an important role in regulating the sensitivity of type I EC towards progestin therapy. Our study opens a new window to explore the mechanisms of progestin resistance.

Nevertheless, the study had several limitations. The sample size was small because only 41 sera data and 20 endometrial tissues were included. More experimental observation and clinical data collection are warranted in the future. In addition, the relationship between progestin resistance and TR function needs further exploration.

## 5. Materials and Methods

### 5.1. Compounds

Medroxyprogesterone acetate (MPA) and nomegestrol acetate (NOMAc) were provided by Xianju Pharmaceutical Co., Ltd. (Taizhou, China). Triiodothyronine (T3) was purchased from Sigma-Aldrich Co., LLC. (St. Louis, MD, USA).

### 5.2. Collection of Human Sera Data and Endometrial Tissues

This study was approved by the Ethics Committee of the Shanghai Institute of Planned Parenthood Research (SIPPR) and the approval No. is PJ2019-10. The data of serum THs were collected from the medical history forms of the patients hospitalized at the Obstetrics and Gynecology Hospital of Fudan University from June 2016 to December 2017, including 41 patients with EC. Adult women diagnosed with EC by primary examination of B-ultrasound and subsequent curettage, hysteroscopy, or surgical pathology were included in the EC group. All 41 patients with EC included were diagnosed for the first time, except for 2 women who were re-diagnosed and 2 women who were found to have EC after surgery for multiple cancers. The serum samples were taken by venipuncture after at least 12 h fasting when they accepted routine hematological examination. None of them had undergone medication treatment. The control group consisted of 67 healthy women from routine physical examinations during the same time. All women had been excluded from malignant tumor-related diseases and thyroid diseases. The thyroid function tests were assessed by extracting peripheral venous blood at fasting status, and serum TSH, free triiodothyronine (FT3), and free thyroxine (FT4) levels were measured using the electrochemiluminescence method.

Moreover, we collected endometrial tissues from the patients hospitalized at the Obstetrics and Gynecology Hospital of Fudan University, including 8 patients with EAH and 12 patients with EC (aged 21–39 years), who underwent hysteroscopy between December 2018 and December 2019. Among them, EAH had 3 progestin-insensitive and 5 progestin-sensitive tissues and EC had 4 progestin-insensitive and 8 progestin-sensitive tissues. All the data were obtained after obtaining informed oral consent.

### 5.3. Cell Cultures

Human RL95-2 EC cell line was purchased from Baili Biotechnology Co., Ltd. (Shanghai, China). The RL95-2 cell line was derived from EC tissues of a 65-year-old white woman. Human KLE EC cell line was derived from the China Type Culture Collection (Wuhan, China). KLE cell line was derived from EC tissue of a 64-year-old woman. Both RL95-2 and KLE cells were cultured in Dulbecco’s Modified Eagle’s Medium/Nutrient Mixture F-12 1:1 (DMEM/F12; Gibco, Carlsbad, CA, USA) media supplemented with 10% FBS (Gibco, New Zealand). All cells were cultured in a humidified incubator at 37 °C with 5% CO_2_ in the air. The culture media were replaced every 2 or 3 days until the cells reached approximately 70–80% confluence, and then the cells were subcultured.

### 5.4. Cell Viability Assays

RL95-2 and KLE cells were seeded in 96-well plates (8000 cells/well), and cells were treated with MPA and NOMAc at concentrations of 1, 3, 10, 30, and 100 µM for 48 h for measurement of half-maximal inhibitory concentration (IC50). In other assays, the cells were treated with T3 (10 or 100 nM), MPA or NOMAc (30 µM), and MPA/NOMAc (30 µM) plus T3 (10 or 100 nM) for 48 h. Control cells were treated with dimethyl sulfoxide (DMSO) (Sigma-Aldrich, St. Louis, MD, USA) and the final concentration in the culture media was 1% (*v*/*v*). According to the manufacturer’s protocol, 10μL CCK-8 solution (Dojindo Laboratories, Kumamoto Prefecture, Kyushu Island, Japan) was added to each well and incubating in a 37 °C incubator for 2 h, and then measured OD value at 450 nm on a microplate reader (BioTek ELX-800). Cell viability was calculated by the formula: cell viability (%) = OD of treatment cells/OD of control cells ×100%. Final results were presented as IC50 with 95% confidence intervals (95% CI), which were calculated from a nonlinear regression model based on log(inhibitor) vs. normalized response/variable slope dose–response curves using GraphPad Prism 8.0 (GraphPad Sofware Inc., La Jolla, CA, USA).

### 5.5. Immunohistochemistry

The biopsied uterine tissues taken from the patients with AC and EAH sensitive or insensitive to the therapy of progestins, including MPA, Mirena^®^, or megestrol (MA), were fixed in formalin and embedded in paraffin. Progestin insensitivity was defined as disease progression at any time during treatment, stable disease after 7 months of treatment, or no complete response (CR) after 10 months of treatment. Other patients who achieved CR within 10 months of treatment were regarded as progestin sensitive. A SABC immunohistochemistry kit (Boster Bio, Wuhan, China) was used to detect the expression of TRα and TRβ in the endometrium following the manufacturer’s instruction. Briefly, the sections were deparaffinized and immersed in antigen retrieval solution (containing Tris 12.1 g, urea 50 g, and ultrapure water 1 L, pH = 9.5) for 10 min at 95 °C. Then, diluted antibodies (TRα 1:100 or TRβ 1:100; Sigma-Aldrich, St. Louis, MD, USA) were dropped onto the tissues and incubated overnight in a humidified chamber at 4 °C. Finally, the sections were stained with DAB working solution (Boster bio, Wuhan, China) for 5–8 min. The double-blind readings were performed by two experienced technicians. The expression of TRα or TRβ in each section was evaluated using histoscores and ratio of positive area. Histoscores were calculated using the following formula: scores of the positive cells multiplied by the grade of staining intensity. The scores of ≤3 points indicated negative expression, and scores of >3 points indicated positive expression. Moreover, the scores of >3 and ≤6 points denoted weakly positive, and scores of >6 denoted strong positive [39]. The number of positive cells in each section was determined by the number of stained cells in 100 cells of 5 random fields under a microscope at 200× magnification. The grade of positive staining intensity was defined as follows: 1 point stood for weak immunostaining and was demonstrated in light yellow color, 2 points for moderate immunostaining and demonstrated in brown color, and 3 points for strong immunostaining and demonstrated in tan color. The criteria for the scores of positive cells were described as follows: no positively stained cells were scored as 0 points, 10–25 stained cells as 1 point, 26–50 stained cells as 2 points, and more than 50 stained cells as 3 points. In addition, the ratio of the positive area in each section was analyzed using Image J 1.48 (Rawak Software, Stuttgart, Germany) and its plug-in IHC Tools. The expression of TRα and TRβ in the sections of progestin-sensitive and progestin-insensitive tissues were then statistically analyzed.

The RL95-2 and KLE cells were seeded onto poly-lysine-coated coverslips (Boster Bio, Wuhan, China) inserted in a 24-well plate at 100,000 cells/well, and cultured in incubator at 37 °C with 5% CO_2_ for 24 h. The coverslips were incubated in 4% paraformaldehyde for 10 min and then immersed in antigen retrieval solution at 95 ℃ for 10 min. The cells were permeabilized with 0.2% Triton X-100 (Sigma-Aldrich, St. Louis, MD, USA), incubated with goat serum blocking solution for 30 min, and then incubated with diluted antibodies targeting TRα (1:40; R&D system, Emeryville, CA, USA) or TRβ (1:200; Abcam, Burlingame, CA, USA) overnight at 4 °C in a humidified chamber; finally, the cells were stained with DAB working solution for 5–8 min. The coverslips were dehydrated and photographed under a microscope (Leica DMi8, Hesse, Wetzlar, Germany)

### 5.6. Small Interfering RNA (siRNA) Transfection

RL95-2 and KLE cells were seeded into six-well culture plates at a density of five million cells/well for 24 h. Transfection experiments were performed when the cell confluence reached 70–90%. Then, 200 pmol of siRNA negative control or siRNA against THRA or THRB (Sangon Biotech, Shanghai, China) was added to Opti MEM (Thermo Scientific, Waltham, MA, USA), followed by Lipo3000 (Invitrogen, Carlsbad, CA, USA) for dilution. The solutions were mixed gently to prepare the Lipo3000 siRNA Transfection Reagent–siRNA complex. Subsequently, the complex was evenly dropped onto RL95-2 and KLE cells and then transfected for 48 h. THRA and THRB siRNA sequences in RL95-2 cells were as follows: THRA: 5′-CAAACACAACAUUCCGCAUUTT-3′; THRB: 5′-GCCUGUGUUGAGAGAAUAGAATT-3′. THRA and THRB siRNA sequences in KLE cells were as follows: THRA: 5′-GCGUAAGCUGAUUGAGCAGAATT-3′; THRB: 5′-GCCUGUGUUGAGAGAAUAGAATT-3′.

### 5.7. RNA Isolation and Quantitative Reverse Transcriptase–Polymerase Chain Reaction Analysis

Trizol (Invitrogen, Carlsbad, CA, USA) was added into RL95-2 and KLE cells or silenced THRA- or THRB-RL95-2 and KLE cells. The cells were collected and incubated in Trizol for 5 min. Chloroform (Sangon Biotech, Shanghai, China) was added to the lysed cells, vigorously shaken and mixed, and then centrifuged at 12,000× *g* for 15 min at 4 °C. The supernatant was discarded, and isopropanol (Sangon Biotech, Shanghai, China) was added to the sediment, shaken, and centrifuged at 7500× *g* for 10 min at 4 °C. After drying, added 15 μL of RNase-free water (Sangon Biotech, Shanghai, China) was added to dissolve, and the concentration on an RNA concentration meter (Thermo Scientific, Waltham, MA, USA) was measured. RNA was reverse transcribed into cDNA on a PCR machine (ABI Veriti) following the instructions of the TAKARA (TAKARA, Tokyo, Japan) reverse transcription kit. The DNA template, primers (THRA, THRB, and ACTB genes) (Sangon Biotech, Shanghai, China), and reagents such as TB Green Premix Ex Taq were mixed, and qPCR was performed following the manufacturer’s instructions (TAKARA, Tokyo, Japan). The primers designed in the experiment are shown in Table 1 (Sangon Biotech, Shanghai, China). The melt curves and Ct values were analyzed using Roche LC480 software (Roche, Basel, Switzerland). The fold change of gene silencing efficiency was calculated using the formula 2^−△△Ct^. The primer sequences used were as follows: ACTB (sense, 5′-CCTGGCACCCAGCACAAT-3′; antisense, 5′-GGGCCGGACTCGTCATAC -3′); THRA (sense, 5′-GATGACACGGAAGTGGCTCTGC-3′; antisense, 5′-AATGTTGTGTTTGCGGTGGTTGAC-3′); and THRB (sense, 5′-CAACTTTTTGGCAAAATCCACC-3′; antisense, 5′-GATGACACGGAAGT GGCTCTGC-3′).

### 5.8. Cell Migration Assay

After silencing THRA or THRB for 48 h, the silenced RL95-2 or KLE cells and negative siRNA control cells were digested and washed with HBSS (Thermo Scientific, Waltham, MA, USA) once or twice, and then resuspended in serum-free media. The densities of the cells were adjusted to 50,000 cells/well. The cell suspension was transferred to a Transwell chamber (Corning, New York, NY, USA) and 600 μL of media containing 10% FBS was added to the lower chamber. Then, 30 μM MPA or NOMAc was added to the cells and incubated at 37 °C in a 5% CO_2_ incubator for 12 h. The control cells were treated with the same volume of DMSO. After 12 h, the residual media in the chamber were discarded, and the chambers were washed twice with PBS (Corning, New York, NY, USA). The remnant cells in the upper chambers were wiped off with a cotton swab and fixed with 4% paraformaldehyde (dissolved in PBS) (Tansoole, Shanghai, China) for 15 min, and then rinsed slowly twice with water for 2 min each time. The cells were infiltrated with 0.1% crystal violet solution (Sangon Biotech, Shanghai, China) for 10 min and then washed with water twice. The staining cells were observed under a microscope at 20× magnification, and five fields of view were randomly selected for photographing and counting. The cells were counted using Image J 1.48 software (Rawak Software, Stuttgart, Germany).

### 5.9. Cell Invasion Assay

Matrigels (Corning, New York, NY, USA) were pre-cooled to 0 °C in advance and then diluted to the concentration of 200–300 μg/mL using serum-free media. Then, 100 μL of Matrigel was added to each transwell chamber. The gel was placed in a 37 °C incubator for 1 h, and the upper liquid media was discarded and used to culture RL95-2 and KLE cells until the logarithmic growth phase. After silencing of THRA or THRB for 48 h, the silenced RL95-2 or KLE cells and negative siRNA control cells were digested. The media were discarded by centrifugation and washed once or twice with HBSS, and then resuspended in serum-free media. The densities of the cells were adjusted to 100,000 cells/well. The procedures for cell inoculation, drug treatment, staining, and cell counting were the same as for the cell migration assay.

### 5.10. Transcriptomic Analysis

RL95-2 cells were transfected with si-THRB or solvent for 48 h and then treated with DMSO, or 30 μM MPA or NOMAc for another 48 h. The cells were harvested and RNA was extracted and checked for quality. After quality inspection, cDNA libraries were constructed. The cells treated with si-Ctrl, si-THRB or si-THRB-DMSO, si-THRB-MPA, or si-THRB-NOMAc, were subjected to high-throughput sequencing after the silencing of THRB using the paired-end sequencing method of the Illumina Hiseq sequencing platform (Juran Biotech, Shanghai, China). For all samples, the raw sequence numbers of known genes were calculated using StringTie software and the expression of known genes was calculated using fragments per kilobase of transcript per million fragments mapped (FPKM). FPKM = total fragments/(mapped reads(millions) × exon length(KB)). The DESeq2 package was used to screen DEGs between different sample groups. The DEGs were screened out by calculating *p* values with Fisher’s exact test. The calculated *p* values were used to determine whether the KEGG functional set in the target genes were significantly enriched or not, and the *p* values were corrected by Benjamini & Hochberg’s multiple tests to obtain a false discovery rate (FDR). The data satisfying |log2FC| ≥ 1 and *p* value ≤ 0.05 were used to screen the DEGs between the two groups. The DEG screened out between si-Ctrl versus si-THRB, si-THRB-DMSO versus si-THRB-MPA, and si-THRB-DMSO versus si-THRB-NOMAc groups were further analyzed using the KEGG signaling pathway. KEGG functional analysis was performed via functional annotation and classification for the pathways in which these genes were involved. The enrichment results were visualized using an online tool (http://www.bioinformatics.com, accessed on 26 April 2022 and 8 October 2022). Final data were from three independent experiments.

### 5.11. Western Blot

The cells silenced with si-THRB for 48 h or the cells were treated with T3 (100 nM), MPA or NOMAc (30 µM), and MPA/NOMAc (30 µM) plus T3 (100 nM) for 48 h, respectively. Control cells were treated with the DMSO. The cells were harvested and suspended in rapid cell-tissue lysis buffer (RIPA) (Invitrogen, Carlsbad, CA, USA) containing 1% protease and phosphatase inhibitors (Thermo Scientific, Waltham, MA, USA). The extracted proteins were boiled at 100 °C for 5 min and then stored at −20 °C. Protein concentration was determined using a BCA protein detection kit (Sangon Biotech, Shanghai, China), and 15 μg of total proteins were electrophoresed and loaded on to 10% SDS-PAGE gels (Sangon Biotech, Shanghai, China) and then transferred to PVDF membranes (Millipore) for 1.5 h. Membranes were blocked with 5% nonfat milk powder (TBST) for 1 h and incubated overnight at 4 °C. Primary antibodies against phospho-Akt (Ser 473) (#9271, 60 kDa), Akt (Pan) (#4691, 60 kDa), phospho-mTOR (Ser2448) (# 5536, 289 kDa), mTOR (# 2983, 289 kDa), phospho-4EBP1 (Ser65) (#9451, 15–20 kDa), 4EBP1 (#9644, 15–20 kDa), phospho-eIF4G (Ser1108) (#2441, 220 kDa), eIF4G (#2469, 220 kDa) and β-actin (#4970, 45 kDa) were diluted at 1:1000 (CST, Danvers, Massachusetts, USA). TRα (R&D Systems, Emeryville, CA, USA, #PP-H2804-00) and TRβ (sigma, St. Louis, MD, USA, #SAB4502969) were diluted at 1:1000; GAPDH (#ab181602, 36 kDa) were diluted at 1:10,000. The PVDF membranes (Millipore, Bedford, MA, USA) were then washed three times with TBST solution and incubated in peroxidase-conjugated goat anti-rabbit IgG (immunoglobulin G) (#7074) with dilution of 1:3000 at room temperature for 1 h. After washing with TBST three times, protein bands were visualized using the ECL SuperSignal West Femto Detection Kit (Thermo Scientific, Waltham, Massachusetts, USA). All antibodies were purchased from Cell Signaling Technology (Beverly). We performed grayscale analysis of the bands using Image Lab 4.0 (Sydney, Australia), and data were analyzed using the method of (p-protein/GAPDH)/(protein/GAPDH).

### 5.12. Statistical Analysis

Data were presented as the mean ± standard error of mean (SEM) of triplicate or three independent experiments, and all statistical analyses were performed with PRISM 8.0 (GraphPad Software, Inc., La Jolla, CA, USA). Multiple comparisons among groups were analyzed using one-way ANOVA (and nonparametric) followed by post-test of Tukey’s or Dunnett’s multiple comparison test. Two-tailed unpaired t-test was used to compare the serum levels of THs between the patients and healthy women, as well as the viability of the cells and the changes of protein expression and transcriptomics prior to and after progestins treatment. Data were considered as statistically significant at *p*-values less than 0.05.

## Figures and Tables

**Figure 1 ijms-23-12517-f001:**
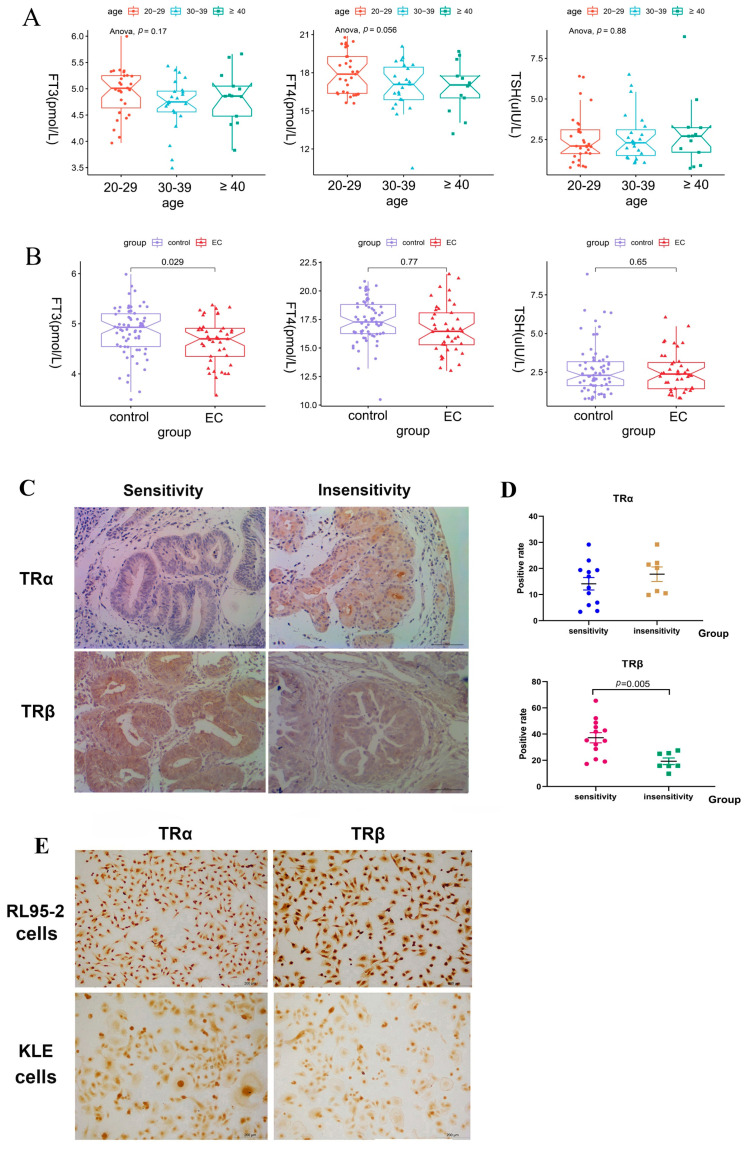
Serum levels of thyroid hormone in healthy women and patients with EC and protein expression of thyroid hormone receptors in the endometrium of patients with EC, endometrial atypical hyperplasia (EAH) and cultured human cells. (**A**) Serum levels of FT3, FT4, and TSH in healthy women of different ages: 20–29 (*n* = 31), 30–39 (*n* = 23), and more than 40 years of age (*n* = 13). (**B**) Serum levels of FT3, FT4, and TSH in the patients with EC (*n* = 41) and healthy women group (*n* = 67). (**C**) Protein expression of TRα in patients with progestin-sensitive (*n* = 12) or progestin-insensitive (*n* = 7) EAH/EC patients, and TRβ in patients with progestin-sensitive (*n* = 13) or progestin-insensitive (*n* = 7) EAH/EC. (**D**) TRα and TRβ were evaluated by the positive rate and are represented by dot plots. Results are presented as mean ± standard error of the mean (SEM). (**E**) Protein expression of TRα and TRβ in RL95-2 and KLE cells.

**Figure 2 ijms-23-12517-f002:**
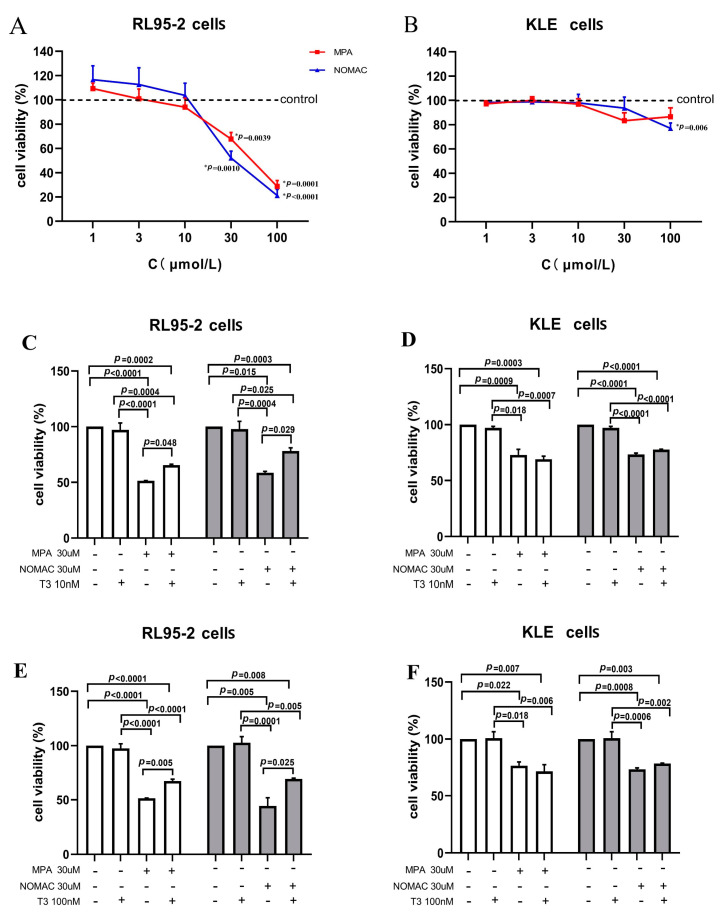
Effects of progestins on the viability of RL95-2 and KLE cells. (**A**,**B**) Cells were treated with MPA or NOMAc at concentrations of 1, 3, 10, 30, and 100 μM for 48 h, respectively. (**C**–**F**) Cells were treated with MPA or NOMAc at a concentration of 30 μM and their combination with 10 and 100 nM T3 for 48 h. The results are presented as mean ± SEM of wells in triplicate from three independent experiments.

**Figure 3 ijms-23-12517-f003:**
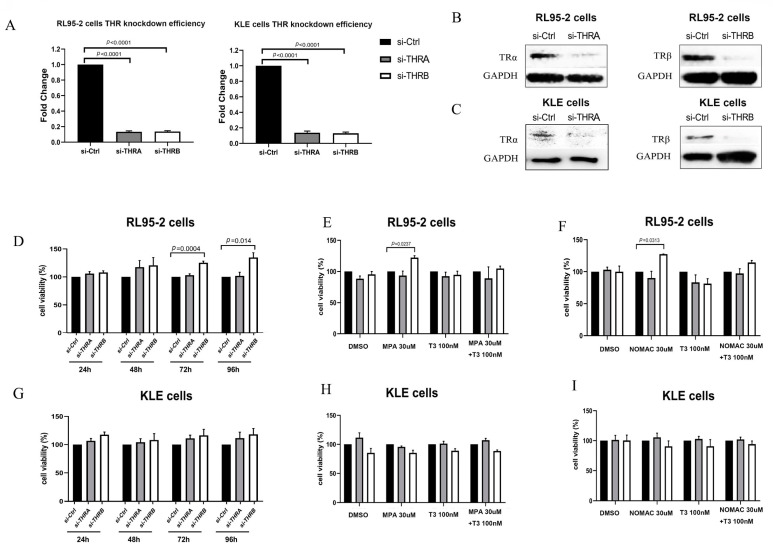
Progestins enhanced the growth of THRB-silenced EC cells. (**A**) THRA or THRB silencing efficiency in RL95-2 and KLE cells. (**B**,**C**) Protein expression of TRα or TRβsilencing in RL95-2 and KLE cells. (**D**,**G**) Cell viability after silencing THRA or THRB in RL95-2 and KLE cells. The cells were treated with si-THRA or si-THRB for 48 h and then cultured in fresh media for 24, 48, 72, and 96 h. (**E**,**F**) RL95-2 or (**H**,**I**) KLE cells were pretreated with si-THRA or si-THRB for 48 h and then treated with 30 μM MPA (**E**,**H**) or NOMAc (**F**,**I**); meanwhile, 100 nM T3 was added for 48 h to examine cell viability. The cell viability was normalized to the control, which was set at 100%. The results are presented as mean ± SEM of three independent experiments. TRα/THRA, thyroid hormone receptor alpha; TRβ/THRB: thyroid hormone receptor beta; si-Ctrl, negative control treated with siRNA solvent; si-THRA, silenced THRA; si-THRB, silenced THRB.

**Figure 4 ijms-23-12517-f004:**
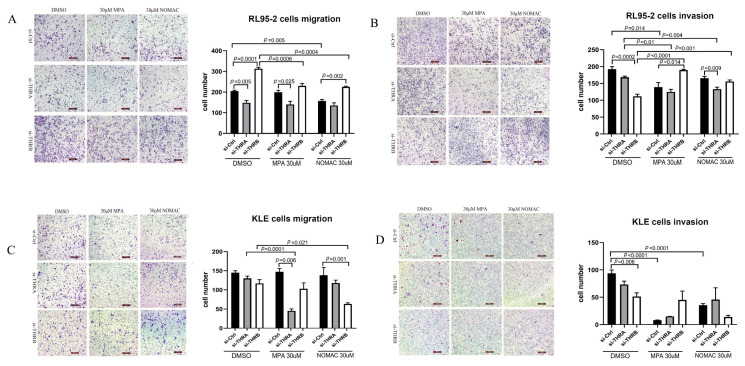
Progestins changed cell migration and invasive abilities in THRB- or THRA-knockdown EC cells. (**A**,**B**) RL95-2 or (**C**,**D**) KLE cells were pretreated with si-THRA or si-THRB for 48 h and then incubated in a polycarbonate filter insert containing DMSO, 30 uM MPA, or NOMAc for 12 h to examine the cell migration (**A**,**C**) and invasion (**B**,**D**). The results are presented as mean ± SEM of three independent culture experiments. si-Ctrl, negative control treated with siRNA solvent; si-THRA, silenced THRA; si-THRB, silenced THRB.

**Figure 5 ijms-23-12517-f005:**
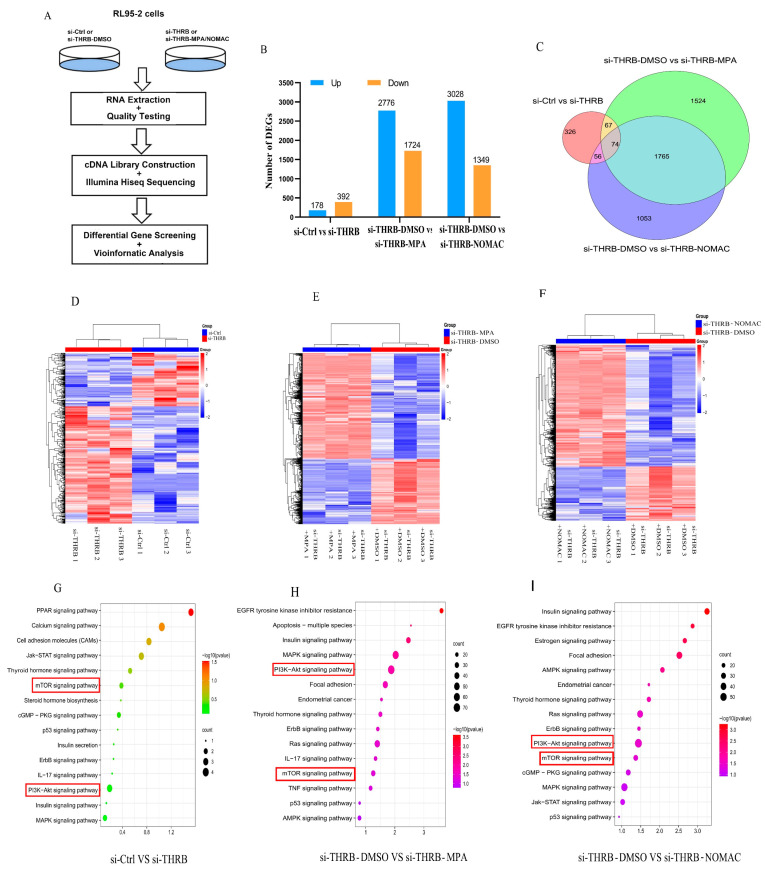
Functional enrichment analysis of DEGs and enriched signaling pathways. (**A**) Transcriptomics flowchart. (**B**,**C**) Column chart and Venn diagram showing the number of DEGs up and down-regulated in the si-Ctrl versus si-THRB, si-THRB-DMSO versus si-THRB-MPA and si-THRB-DMSO versus si-THRB-NOMAC groups. (**D**) Heatmap of DEGs between si-Ctrl-RL95-2 and si-THRB-RL95-2 cells. (**E**) Heatmap of DEGs between DMSO and MPA treatments in THRB-silenced RL95-2 cells. (**F**) Heatmap of DEGs between DMSO and NOMAc treatments in THRB-silenced RL95-2 cells. (**G**) KEGG enrichment analysis of DEGs between si-Ctrl-RL95-2 and si-THRB-RL95-2 cells. (**H**) KEGG enrichment analysis of DEGs between si-THRB-DMSO-RL95-2 and si-THRB-MPA-RL95-2 cells. (**I**) KEGG enrichment analysis of DEGs between si-THRB-DMSO-RL95-2 and si-THRB-NOMAC-RL95-2 cells. The results are presented as mean ± SEM of three independent experiments. THRB, thyroid hormone receptor beta; si-Ctrl, negative control treated with siRNA solvent; si-THRB, silenced THRB; MPA, medroxyprogesterone acetate; NOMAc, nomegestrol acetate; DEGs, differentially expressed genes.

**Figure 6 ijms-23-12517-f006:**
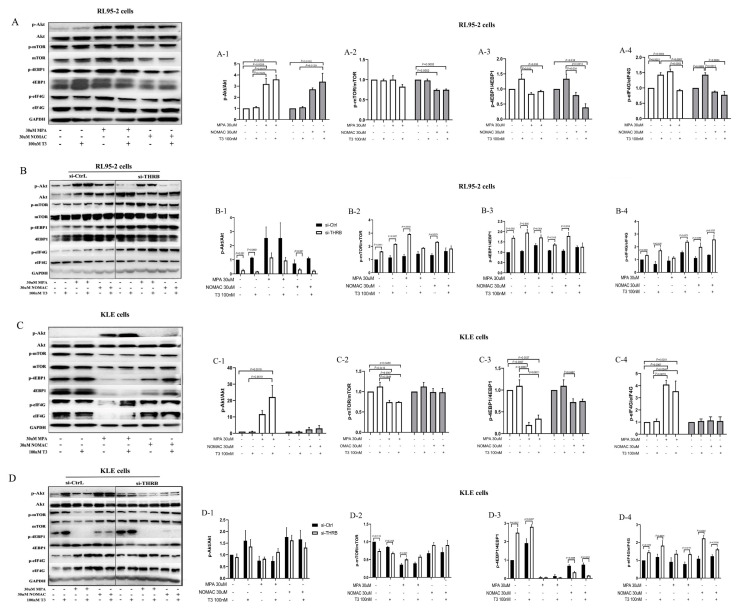
Effects of MPA and NOMAc on protein levels of Akt/ mTOR/ 4EBP1/eIF4G in EC cells. (**A-1,A-2,A-3,A-4**) The expression levels of p-Akt/Akt, p-mTOR/mTOR, p-4EBP1/4EBP1 and p-eIF4G/eIF4G in wild-type RL95-2 cells. (**B-1,B-2,B-3,B-4**) The expression levels of p-Akt/Akt, p-mTOR/mTOR, p-4EBP1/4EBP1 and p-eIF4G/eIF4G in THRB-silenced RL95-2 cells. (**C-1,C-2,C-3,C-4**) The expression levels of p-Akt/Akt, p-mTOR/mTOR, p-4EBP1/4EBP1 and p-eIF4G/eIF4G in wild-type KLE cells. (**D-1,D-2,D-3,D-4**) The expression levels of p-Akt/Akt, p-mTOR/mTOR, p-4EBP1/4EBP1 and p-eIF4G/eIF4G in THRB-silenced KLE cells. The results are presented as mean ± SEM from three independent experiments. THRB, thyroid hormone receptor beta; si-Ctrl, negative control without THRB silencing; si-THRB, THRB silencing; T3, triiodothyronine; MPA, medroxyprogesterone acetate; NOMAc, nomegestrol acetate; Akt, phosphorylated protein kinase B; mTOR, mammalian target of rapamycin; 4EBP1, 4e-binding protein 1; eIF4G, eukaryotic translation initiation factor 4G.

**Table 1 ijms-23-12517-t001:** Antiproliferative activity of MPA and NOMAc on RL95-2 and KLE cells after treatment for 48 h.

Drugs	Inhibitory Potency IC50 (95%CI) μM	
	RL95-2 Cell Lines	KLE Cell Lines
MPA	52.25 (38.95–71.70)	/
NOMAc	36.81 (23.23–65.47)	281.2 (194.4 to 489.2)

Data were calculated from wells in triplicate from three independent assays and presented as IC50 (95% confidence interval) values (μM). / means that the corresponding IC50 value was greater than 400 μM or could not be calculated because the viability of cells did not reach 50% of the maximum. NOMAc: nomegestrol acetate, MPA: medroxyprogesterone acetate.

## Data Availability

Not applicable.

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
