# Peer review of "Thyroid Hormone Receptor β Knockdown Reduces Suppression of Progestins by Activating the mTOR Pathway in Endometrial Cancer Cells"

_ijms, 2022, doi:10.3390/ijms232012517_

Round 1

Reviewer 1 Report

This manuscript examines the interplay of thyroid hormone receptor and thyroid hormone supplementation with progestin-mediated modulation of the mTOR pathway in endometrial cancer cells. While the results could have interesting implications for treatment of this type of cancer, conclusions seem preliminary. Major concerns that need to be addressed are as follows:

1. Fig. 1B reports a significant difference in serum FT3 levels between healthy women and EC patients; however, the difference in pmol/L is negligible (is this slight change of physiological significance?; the authors need to put these values in context) and is based on an insufficiently small sample size (healthy, N=67, EC N=41). Given that thyroid dysfunction occurs only in about 8.2% of patients with EC, a much larger sample size would be required.

2. The authors need to provide more information about the two EC cells lines used, given that they differ in responses. What are their different characteristics and what is already known about their behavior in the literature?

3. For valid studies of 100 nM T3 supplementation, cells must be cultured in charcoal/dextran-stripped FBS. It is well known that standard FBS contains physiological (but variable) levels of T3, so the authors are comparing "some amount of T3" (not 0 nM) against "100 nM." Also, 100 nM is not considered physiological. It would be of interest to determine what effects there might be at lower levels (e.g. 2.5 to 10 nM).

4. It is not sufficient to show knockdown of mRNA levels only, given that nuclear receptors may have slow turnover rates in the cell. The authors should include a western blot analysis. This would also serve to validate the antibodies that were used for immunohistochemistry. Are they truly specific for the two subtypes of TR, TRB1 and TRa1 (in general, commercially-available antibodies are relatively non-specific)? Do they recognize the correct-sized bands on a Western? Fig. 2A would need to be quantified to make the statement that expression is "stronger" in RL95-2 cells.

5. In Fig. 2E-F, the average is the same for NOMAc  and NOMAc + T3, but the SEM is greater for the latter. Can the authors provide an explanation for why these results have high variability and hence are not statistically significant?

6. In the figure legend of Fig 2, for ease of understanding, explain that cell viability is normalized to the control, which is set at 100%.

7. The authors should provide better rationale for why they chose to only focus on Akt and mTOR signaling, rather than also exploring other pathways that came up in the DEG analysis (Fig. 3), such as MAPK and Ras. They have not ruled out indirect effects on these other pathways.

8. The authors need to explain their seemingly contradictory result that supplementation with T3 has a significant effect on the pathway when THRB is knocked down. Is this because, in fact, knockdown at the protein level was insufficient, or is it because TRalpha is also of importance?

9. Some lanes of the Western blots shown in Fig. 4 appear oversaturated, calling into question the accuracy of quantification. The authors need to provide an explanation in the methods section for how blots were quantified.

10. Fig. 4. It is not possible to do an appropriate statistical analysis on n=2 (see figure legend regarding KLE data). A third independent replicate is required.

Reviewer 2 Report

This is a very well-focused work investigating the modulatory effects of thyroid hormone signaling in the promotion/suppression of endometrial cancer cell proliferation in the context of treatment with progestins. Using an in vitro approach and silencing/knockdown  of THRs, the authors report that blocking thyroid signaling through the TH receptor beta detracts significantly from the beneficial effects of progestin treatment on decreasing cancer cell viability, migration and invasion. They also show that T3 treatment may also work in the same manner when THRB is silenced.  Using a transcriptomic approach and Western analyses, the authors identified alterations in the mTOR signaling pathway as potentially contributing to the effects associated with manipulating T3 signaling.

The experiments are well performed, and the results are novel and provide valuable information about the roles of thyroid hormone signaling in cancer progression in general.

I have the following comments to improve the manuscript:

The abstract is poorly written and structured: the experiments and conclusion are not clear.

Line 88: what is the EC status of these patients? Were they just diagnosed? Were they already under treatment? There is not information in the Methods section about the patients from whom the serum samples originated.

The tissue origin of the cell lines used in the study should be clearly stated in the methods.

Figure 2B: Can the authors comment, based on the CT values, on the apparent relative abundance of each of the receptors in the cell lines used?

100 nM T3 is a very high dose for cultured cells. Have the authors tried a lower dose?

The font size in many figures is excessively small.

In the transcriptomics experiment, it is strange why the “thyroid hormone signaling” pathway is enriched in progestin treated samples, but not in control, THRB-silenced samples. Do the authors have an explanation?

A Venn diagram(s) to show the overlap of DEGs shown in Figure 3B will be helpful.

Supplementary data sheets listing some of the genes in figure 3B will also be valuable. The authors do not highlight or discuss any specific, prominent DEGs, especially ones that may be also involved in T3 signaling or that may be known T3 targets.  It is not clearly justified why other pathways/genes were not selected for follow-up.

Lines 294-8:  A causative effect of low T3 cannot be inferred. In most hypothyroid cases, it is primarily T4 that is low, not T3. A more likely explanation of the T3 results is the non-thyroidal illness syndrome.

Line 306-7: this statement does not seem correct. It is not THRB signaling that is boosting cancer cell proliferation, it is THRB knockdown that does it. So the conclusion is that THRB is really having a suppressing role in cancer cell proliferation and, when THRB is knocked down, the suppressing effect is relieved.

Along the same lines, If T3 treatment has an effect -any effect of the ones described- in THRB-silenced cells, wouldn’t its action be mediated by the THRA? This possibility or any alternatives are not discussed.

Editing suggestions:

The English is generally poor. In many instances, it does compromise scientific clarity. The assistance of a native English editor is strongly advised.

Round 2

Reviewer 1 Report

The authors have addressed my concerns in a satisfactory way in their revised manuscript.

The manuscript should be proofread again carefully. For example, there is an extra "o" in line 52, the spelling of "Sensitivity" and "Insensitivity" in the labeling of Figure 1C needs to be corrected, and on line 132, the spelling of "dependent" needs to be corrected.

Author Response

Thanks a lot for your kind reminder. We have carefully checked out and proofread  the manuscript and corrected all the errors including you mentioned.

Reviewer 2 Report

The revisions have significantly improved the manuscript.

Still I think the authors should provide some Supplementary data including lists of all or selected differentially expressed genes in figure 5, specially those related to thyroid hormone, PI3K and mTOR signaling.

Author Response

Thanks for your comment. We have added the lists of DEGs and heatmaps correlated to the pathways in Table S1, S2 and S3 and Figure S1, respectively, as a supplement to the data. Please see line 256-263, Figure S1 and Table S1, S2 and S3.